# Spatial and Temporal Genetic Diversity of the Peach Potato Aphid *Myzus persicae* (Sulzer) in Tunisia

**DOI:** 10.3390/insects10100330

**Published:** 2019-10-01

**Authors:** Amen Hlaoui, Sonia Boukhris-Bouhachem, Daniela A. Sepúlveda, Margarita C.G. Correa, Lucía M. Briones, Rebha Souissi, Christian C. Figueroa

**Affiliations:** 1Laboratoire de Protection des Végétaux, Institut National de Recherche Agronomique de Tunisia INRAT, Rue Hédi Karray, Ariana 2049, Tunisia; hlaoui_amen@hotmail.com (A.H.); bouhachems@gmail.com (S.B.-B.); srebha@yahoo.com (R.S.); 2Département Santé Végétale et Environnement, Institut National Agronomique de Tunisia INAT, Université de Carthage, 43 Avenue Charles Nicolle, Cité Mahrajène Tunis 1082, Tunisia; 3Centre for Molecular and Functional Ecology in Agroecosystems, Universidad de Talca, Talca 3460000, Chile; dani.sepulveda.14@gmail.com (D.A.S.); maggiecorrea@gmail.com (M.C.G.C.); lbriones@utalca.cl (L.M.B.); 4Facultad de Ciencias Agrarias, Universidad de Talca, Talca 3460000, Chile; 5Institut National de la Recherche Agronomique INRA, CNRS, ISA, Université Côte d’Azur, 06903 Sophia Antipolis, France; 6Instituto de Ciencias Biológicas, Universidad de Talca, Talca 3460000, Chile

**Keywords:** peach potato aphid, *Myzus persicae*, Tunisia, genetic diversity, microsatellites, population structure

## Abstract

The peach potato aphid, *Myzus persicae* (Sulzer), is a worldwide pest of many crops, and the most important aphid pest of peach and potato crops in Tunisia, mainly due to virus transmission, for which insecticides are frequently applied. We studied the genetic structure of *M. persicae* populations in Tunisia, in order to further our understanding of the biotic and abiotic factors shaping populations and to predict their evolutionary responses to the present management practices. We monitored peach orchards and seed potato crops in different seasons and regions from 2011–2013 and in 2016 (19 populations), assessing the genetic diversity of *M. persicae* at six microsatellite loci. Temporal and spatial changes in the frequency and distribution of 397 genotypes in 548 sampled aphids were studied. Only 37 genotypes were found more than once (clonal amplification), as most genotypes were found only once (91.60% in peach; 88.73% in potato crops). A similarly high genetic diversity was observed in aphids sampled from peach (G/N = 0.76; Ho = 0.617) and potato (G/N = 0.70; Ho = 0.641). Only a weak genetic differentiation among populations was found, mainly between geographic locations. Clustering analysis revealed genotypes to be grouped mainly according to host plant. The availability of the primary host, high proportion of unique genotypes, high genetic diversity and lack of structuring suggest that the aphid reproduces mainly through cyclical parthenogenesis in Tunisia. On the other hand, we provide a farm-scale study that shows how easily *M. persicae* can colonize different areas and hosts, which may have important implications in relation to plant virus vectoring.

## 1. Introduction

In agroecosystems, insect pest populations can rapidly evolve to the latest pest management practices and global changes [1]. Understanding the genetic features of pest populations is a requirement to design novel pest control tactics based on anticipating their evolutionary responses [2]. The monitoring of pests allows for the collection of data on the distribution of genetic diversity and the factors that shape the spatial and temporal structuring of their populations.

Aphids (Hemiptera: Aphididae) can pose a major threat to food crop safety due to direct physical damage and vectoring viral plant diseases [3]. Aphids have colonized almost all agroecosystems worldwide, and they can display polyphenisms in response to environmental changes, including winged/wingless and sexual/asexual individuals in the same population [4]. Aphid pests can successfully respond to agricultural practices (e.g., insecticide resistance, high adaptive plasticity to resistant plants and natural enemies), and can rapidly build up large populations due to parthenogenetic reproduction, thus easily reaching the economic threshold [5].

The peach potato aphid, *Myzus persicae* (Sulzer), is one of the most exceptional aphid pest species globally. Originating in Asia, this aphid has a cosmopolitan distribution, being found anywhere crops are cultivated [6]. The aphid is a highly polyphagous insect herbivore feeding on a wide range of hosts belonging to more than 400 species from 50 plant families, including agro-industrial crops (e.g., potato, tomato, tobacco), horticultural crops (e.g., sweet pepper, cabbage), and stone fruits (e.g., peach, cherry) [3]. Furthermore, the aphid is a highly efficient vector of phytopathogenic viruses like potato leafroll virus (PLRV), potato virus S (PVS), and potato virus Y (PVY), which represent the most serious threat to potato production worldwide, and plum pox virus (PPV) that produces Sharka disease in peach trees [7]. Populations of the peach potato aphid are primarily controlled by the application of synthetic insecticides, but this has led to the rapid evolution of several mechanisms of resistance [8].

The reproduction mode, host plant availability and climate are all drivers shaping aphid population structures and allowing aphids to spread and colonize extensive geographic areas [4]. The peach potato aphid displays complex and variable reproductive strategies.

It is a host-alternating aphid with populations comprising different genotypes produced in a single sexual generation on the primary woody host *Prunus persica* (L.) Batsch. In the autumn, pre-sexual females (gynoparae) give rise to sexual females (oviparae) that produce cold-resistant eggs after mating with winged males. In the spring, fundatrix females (the first female asexual morphs within a lineage) emerge from hatched eggs and multiply by parthenogenesis during spring and summer on secondary herbaceous hosts (i.e., cyclical parthenogenesis, CP) [9,10]. When the primary host is absent, the aphid may still reproduce, but exclusively by parthenogenesis on secondary hosts all-year-round. In this case, populations will comprise a few overrepresented genotypes, which generate a strong biased representation of some aphid genotypes due to differential clonal amplification [11]. In addition, some genotypes from CP populations may lose their ability to reproduce sexually, either totally (obligate parthenogenesis, OP) or partially (functional parthenogenesis, FP) [12]. While the sexual phase generates frost-resistant eggs that allow aphids to overwinter in cold climates, asexual reproduction allows the retention of those genetic architectures that successfully reproduce in temperate regions and which may favor aphid invasions [13]. Hence, the coexistence of sexual and asexual populations of *M. persicae* and their clonal composition and genetic diversity depend on (i) the availability and abundance of primary and secondary hosts, (ii) how harsh or mild the winters are, and (iii) the intensity of insecticide applications [2].

In Tunisia, *M. persicae* is the most important aphid pest [14,15], particularly of peach and potato crops [16]. It is the main vector of potyvirus (PVY) and polerovirus (PLRV), the most damaging plant viruses for seed production and tuber health in Tunisia, with transmission efficiency rates of over 95% for the PVY^NTN^ strain and PLRV [17,18,19,20]. Thus, frequent applications of neonicotinoid, pyrethroid, organophosphate, and carbamate insecticides are needed to control *M. persicae* populations in spring and summer, while mineral oil is used to control aphids in winter.

Despite the relevance of *M. persicae* as a serious pest and its significant negative agronomical impacts, few studies have been conducted in Tunisia on the biology of this aphid, most of them focused on virus dissemination. This aphid has been reported as CP in Tunisia, with male and female sexual morphs, overwintering eggs, and winged and wingless parthenogenetic morphs being described [21]. But, little is known about the genetic features of *M. persicae* populations in Tunisia, although a recent work aimed at studying variation in target-site insecticide resistance described a high clonal diversity and population differentiation among three geographic zones, but on a restricted sub-sample of aphids [22].

In the present study, we investigated the genetic characteristics of *M. persicae* populations in Tunisia aimed at understanding the role of the host plant, reproduction mode and climate variation on the frequency persistence, and distribution of aphid genotypes. Using a larger sample (N = 548) obtained from the monitoring of peach orchards and seed potato crops from different seasons and regions, we assessed the genetic diversity and differentiation within and among populations of *M. persicae* at six microsatellite loci, comparing our results with those previously reported for Tunisia and other countries. Our results are discussed in terms of the spatial temporal dynamics of aphid genotypes and its utility for pest management.

## 2. Materials and Methods

### 2.1. Sampling

*Myzus persicae* were sampled on peach orchards *Prunus persica* (L.) and potato crops (*Solanum tuberosum* L.) located in Cap Bon (36° 50′ 34′′ N, 10° 36′ 44′′ E), Jendouba (36° 33′ 42′′ N, 8° 56′ 40′′ E) and Kairouan (33° 39′ 50′′ N, 9° 59′ 10′′ E). Sampling areas were separated by about the same distance (Cap Bon and Kairouan are 117 km apart; Kairouan and Jendouba are 135 km apart; Jendouba and Cap Bon are 138 km apart (Figure 1) and selected due to the abundance of peach orchards and potato crops. Crops within each location were separated by around 1 to 20 km. Peach has been historically cultivated in these particular regions of Tunisia due to the favorable edaphoclimatic conditions, while seed potato production is cultivated since the implementation of the National Program for Seed Production and Multiplication (Groupement Interprofessionnel des Légumes, GIL). Regarding the climate conditions, Cap Bon is a coastal zone characterized by sub-humid climate (mean T°: min = 15.4 °C, max = 24.3 °C), whereas Jendouba (mean T°: min = 12.5 °C, max = 25.5 °C) and Kairouan (mean T°: min = 15.6 °C, max = 27.5 °C) are located in the continental zone with sub-humid and arid climates, respectively. Climatic data were obtained from the Institut National de la Météorologie in Tunisia. Peach orchards and potato crops were conventionally managed to apply neonicotinoid alternated with pyrethroid insecticides.

Sampling was done on randomly chosen plants by taking one single wingless individual aphid per plant to limit the chance of re-sampling individuals from the same asexual lineage, particularly in spring. The sampling was conducted in 2011, 2012, 2013 and 2016 during the autumn, winter and spring seasons (Table 1). Hence, in total 618 individuals were collected and stored in 95% ethanol for further determination and DNA isolation. All samples were positively identified as *M. persicae* under a binocular microscope following taxonomic keys [3]. Based on the host plant, season, locality and year of collection, the samples were grouped in 19 populations (Table 1).

### 2.2. Genotyping

Genomic DNA was extracted from each sampled individual aphid following the “salting out” protocol [23]. After checking the DNA quality and quantity in a Nanodrop (Nanodrop Technologies, Wilmington, DE, USA) spectrophotometer, the sample size was reduced to 548 individuals. Six microsatellite loci previously described for *Myzus persicae* were used for genotyping each individual; five loci (Myz9, M37, M40, M49 and M63) are autosomal, while M86 is X-linked [24,25]. PCR reactions were performed using the M13 universal primer (−21) labeled with fluorescent FAM or VIC at the 5′-end of the forward primer as described previously [26]. Each amplification was conducted in 15 µL reaction volume containing 1× Mg^+2^-free reaction buffer, 25 mM MgCl_2_, 10 µM dNTPs, 1 µM each forward and reverse primers, 1 µM primer M13, 0.5U Platinum^®^ Taq DNA polymerase (Invitrogen, Carlsbad, CA, USA), and 20 ng/μL of total DNA, all in sterile nano-pure water. PCR reactions were carried out on a MyCycler^®^ thermal cycler (Bio-Rad, Hercules, CA, USA) using the following steps: An initial denaturation at 94 °C for 5 min followed by 4 touchdown cycles consisting of 30 s of denaturation (94 °C), 30 s of annealing (62 °C; 61 °C; 59 °C; and 57 °C each cycle) and 45 s of elongation. Subsequently, 26 cycles with the same steps and times described above but changing the annealing temperature at 55 °C were performed. Finally, 8 cycles using annealing temperature at 53 °C followed by 10 min of elongation at 72 °C completed the amplification. Positive DNA amplifications were checked following electrophoresis in 2.0% agarose gel. Allele size determinations were conducted through automated sequencing at Macrogen, Inc (Seoul, Korea). The allele configuration for each individual was obtained using the software GeneMarker^®^ (SoftGenetics, State College, PA, USA). All microsatellite data were checked for null alleles and/or technical artifacts using the software Micro-Checker 2.2.3 [27].

### 2.3. Data Analysis

#### 2.3.1. Multilocus Genotypes (MLGs)

To determine the clonal diversity and genetic composition of *M. persicae* populations in Tunisia, aphid samples were analyzed as multilocus genotypes or MLGs (i.e., the genotype resulting from the allele combination at all six microsatellite loci amplified). This approach assumes that individuals with the same genotype have a good chance of descent from a genetically identical asexual ancestor or clone, but with modifications due to random mutations within asexual lineages [28]. The correct assignation of each sample carrying an identical allele composition to a given MLG was done using the software GenClone v2.0 [29].

Due to clonal amplification during the asexual phase, aphids within a population can be comparatively more genetically related than individuals in populations of any other diploid organism. Because of this, data analyses were conducted considering one single copy per MLG to avoid the over-representation of some asexual lineages, as well as on the whole sample [28,30]. Also, the population genetic analyses were based on genotypic frequencies rather than allelic frequencies, using the Hardy Weinberg equilibrium (HWE) only to compute the expected frequencies for each MLG [31].

To estimate the genetic relatedness among MLGs, a distance matrix was built based on the allele shared distance (DAS) computed in the software Populations v1.2.32 [32]. To graphically show the similarities among MLGs and to identify clusters of highly similar genotypes that may have evolved from the same asexual ancestor, a distance method neighbor-joining tree was built in the software POPULATIONS and visualized using FigTree v1.4.3 [33].

#### 2.3.2. Genetic Diversity in Populations

The genetic diversity was quantified for the whole sample and for each population using different indexes available in the packages GenClone v2.0 [29], Arlequin v3.5.2.2 [34] and Fstat v2.9.3.2 [35]. This allowed calculation of the following parameters: *First*, the clonal diversity (G/N), the gross genetic diversity standardized by sample size, where G is the number of MLGs and N is the number of genotyped individuals; *second*, the ratio between unique (U; those MLGs observed only once in a sample) and multicopy MLGs (M; those MLGs observed more than once in a sample) (U/M), a raw value of the genotypic richness by population; *third*, the Shannon Weaver (H) and Simpson (S) diversity indexes and their evenness, the latter giving a proportional measure to the actual genotypic richness within populations; *fourth*, the number of alleles per locus (Na); *fifth,* the mean standardized allelic richness over loci for each population (A); sixth, the expected (He) and observed (Ho) heterozygosity; *seventh*, the linkage disequilibrium (LD) expressed as the proportion of the number of linked pairs of loci/the total number of possible pairwise comparisons and departures from the HWE (*F_IS_*) which are estimators of genetic recombination within populations. Altogether, these genetic indexes are good estimators of the level of asexuality/sexuality in aphid populations, which in combination with information on management practices or climate, contribute to a better understanding of the dynamics of aphid populations regarding the predominance of asexual genotypes, their relative success overtime on different hosts, their spread in space, and the chance they may be replaced by their sexual counterparts [13].

#### 2.3.3. Standard Population Genetic Analysis

The heterogeneous distribution of the genetic diversity among the 19 studied populations was assessed using Arlequin v3.5.2.2 [34] and tested for significance of multiple pairwise comparisons using Fisher’s method. Meanwhile, the genetic differentiation among populations due to genotypic frequencies was assessed by calculating the fixation indexes (*F*_ST_) considering one single copy per MLG. The assumptions of the island-model can be transgressed when the use of *F*_ST_ is limited to the quantification of the genetic differences among aphid populations [36]. Hence, *F*-statistics were computed according to Weir and Cockerham [37] with bootstrapping of 1000 replicates [38]. Furthermore, the proportion of the genetic differentiation among populations due to molecular differences was assessed for the 19 populations (whole sample or one single individual per genotype) and their covariance with spatial and temporal factors (i.e., sampling site, year, season and host plant) determined using an hierarchical analysis of molecular variance (AMOVAs), including among groups and among individuals within groups, as well as within individuals of the same sources of variation cited above [39]. Lastly, a migration test was assessed in Genepop v4.0 [40] based on computing the average number of private alleles p(1) and estimating the number of migrants as Nm, where N is the population size and m is the proportion of migrants [41].

#### 2.3.4. Bayesian Population Genetic Analysis

A Bayesian clustering approach was used to assign MLGs to genetic groups and then assess the level of genetic differentiation in the whole dataset independently of the origin of genotypes (e.g., sampling site, year, season or host plant), as implemented in the software Structure v2.3.4 [42]. This approach minimizes deviations from HWE and LD, thereby allowing determination of the number of clusters that best represented data differentiation. The estimation of the number of subpopulations was computed considering (i) a single large population using one single copy per MLG (N = 397), (ii) 19 distinct populations using one single copy per MLG in each population (N = 419), and (iii) 19 distinct populations using all MLGs copies (N = 548). The analysis was run using an admixture model of ancestry (i.e., each individual is represented by a fraction of its genome coming from some of the K hypothetical populations in the sample) with correlated allele frequencies. The numbers of clusters (K) were set from 1 to 19 (19 the maximum number of populations) and a total of 10 replicate runs were performed for each value of K. Each run comprised 1,000,000 Markov chain Monte Carlo (MCMC) iterations after a burn-in period of 200,000 iterations. We used the ΔK approach [43] implemented in the software Structure Harvester [44]. Although this clustering algorithm assumes panmixia, the approach is still robust enough when some assumptions are violated due to the asexual reproduction mode of aphids [45]. When multimodality was observed over Structure runs, the most frequent clustering pattern for a given K-value was identified using the package CLUMPP [46] and the results were plotted using Distruct v1.1 [47].

## 3. Results

### 3.1. Microsatellite Polymorphism in M. Persicae Populations from Tunisia

The six microsatellite loci were highly polymorphic, producing a total of 92 different alleles, averaging 15 alleles per locus (see Appendix A for the allelic combinations of each MLG), and ranging from 10 to 23 alleles for loci M40 and Myz9, respectively (Appendix A). Overall, a deficit of heterozygosity was detected (*F*_IS_ = 0.158), probably influenced by the high homozygosity observed for locus M49 (*F*_IS_ = 0.641). This is doubtless the consequence of null-alleles as revealed by the Micro-Checker package (data not shown). Positive *F*_IS_ either a heterozygote deficiency was detected when the analysis was performed on the whole sample (N = 548) or considering one single copy per genotype (N = 397), although slightly significant differences were observed for locus Myz9 (Appendix A).

### 3.2. Diversity and Distribution of Multilocus Genotypes (MLGs)

A total of 397 MLGs of *M. persicae* were identified from 548 aphids genotyped (G/N = 0.724). On peach, 262 MLGs were detected of the 344 individuals sampled (G/N = 0.762), while 142 MLGs were found in 204 individuals collected from potato seed crops (G/N = 0.696) (Table 2). Less than 10% of genotypes were sampled more than once (hereafter multicopy or mMLGs), which contrasts with the high diversity of unique genotypes (hereafter uMLGs) (Table 2). Both samples from peach and potato showed a diversity of uMLGs close to 90% (Table 2). Seven mMLGs (G6, G77, G159, G278, G368, G369 and G370) were shared between hosts (Table 3). Four mMLGs (G6, G247, G370, and G331) were very common and constituted about 16% of the whole sample (Table 3).

Regarding the distribution of some mMLGs according to host plant, it is noteworthy that G331 was the most frequent mMLG on potato crops. Furthermore, G331 was the only genotype present during the entire sampling period (Table 3, Figure 2), with 6 copies in Cap Bon and 10 copies in Jendouba, followed by G334 with 1 copy in Cap Bon and 8 copies in Jendouba. Other interesting genotypes were G383, with 5 copies in Cap Bon and G214, with 2 copies in Cap Bon and 3 in Jendouba (Table 3, Figure 2). Other genotypes exhibited frequencies less than 5 copies. Overall, of the 37 mMLGs, 17 mMLGs were present only in the peach populations, which is a higher number than the 13 genotypes present only in potatoes.

For *M. persicae* populations on peach, genotypes G6 (8.4%) and G247 (6.4%) were the most represented. These mMLGs shared the same season and locality (spring, Cap Bon) but in different years (2016 and 2013, respectively for G6 and G247) (Table 3, Figure 2).

When the temporal prevalence of *M. persicae* genotypes was studied, four mMLGs (G370, G159, G214 and G278) were found to be the commonest in 2011 and 2016, although their frequencies were variable among hosts, seasons and localities (Figure 2). In contrast, genotype G368 showed the same frequency and was present from 2011 to 2013 at about the same frequency, but with changes in its distribution according to the host, season and location. G330 was found in both 2011 and 2012 at an invariant frequency according to the host, season and location (1 aphid, potato, Jendouba, winter). Other genotypes were repeated only twice, and they were specific for a host, a season or locality (Figure 2).

### 3.3. Genetic Diversity and Structure Within Populations

The clonal diversity measured as G/N in Tunisian populations of *M. persicae* was similarly high to those observed in France and China [10,48] and finds good support in other indexes of diversity we computed, thus suggesting the peach potato aphid in Tunisia exhibits a high genetic diversity regardless of origin or the sampling size (Table 2). The highest diversity was recorded on samples from peach (G/N = 0.762; H index = 5.196; He = 0.775), particularly during winter and autumn (pop5, pop7, pop12, and pop14) from Cap Bon and Jendouba. In potato crops (G/N = 0.696; H index = 4.533; He = 0.798), two populations (pop4 and pop18) showed the highest indexes of diversity in Cap Bon and Kairouan during autumn and winter, respectively (Table 2).

Most *M. persicae* populations in Tunisia significantly deviated from HWE, mainly due to a deficit of heterozygotes (Table 2), a situation expected for aphid populations, as individuals originated via CP involving sexual reintroduction as the predominant reproductive mode during the winter. This observation is further supported by global test of linkage disequilibrium showing that the number of linked loci for populations on peach were less than potato, as well as when populations of each hosts were analyzed separately. Thus within peach population, absence of significant linkage was found for pop19 sampled in Kairouan during winter season, as well as a few linked loci (1 pair of loci was linked out of 15 tests of pairwise comparison) found for pop5 and 7 sampled in the Cap Bon region during autumn and winter, respectively. However, the highest number of loci in linkage disequilibrium was found for pop2 (13/15) and pop6 (10/15) sampled from the same locality Cap bon during the spring and winter season, respectively. The only exceptions were found for pop2 and pop9 which were sampled during spring on peach in Cap Bon and Jendouba and showed negative inbreeding coefficients (Table 2), individuals that were probably the direct offspring of asexual ancestors.

### 3.4. Genetic Differentiation Among Populations Based on the Frequency of MLGs

The spatial and temporal genetic differentiation was analyzed using several approaches. The global differentiation analysis based on all populations showed a weak genetic differentiation among populations, both when considering the whole sample (*F*_ST_ = 0.072, *p* < 0.05) or a single copy per MLG (*F*_ST_ = 0.038, *p* < 0.05). Pairwise comparisons of *F*_ST_ values between each population are shown in (Table 4). Given the unbalanced sampling sizes for some populations (e.g., pop15 and pop8 have 5 and 7 individuals, respectively), we performed comparisons on the distribution of the genetic diversity by grouping samples according to the host plant (peach and potato), locality (Cap Bon, Jendouba and Kairouan), sampling year (2011, 2012, 2013 and 2016) and season (winter, spring and autumn) which allowed evaluation of the impact of each factor and their interactions on the genetic structuring of populations.

When aphid populations were grouped according to their geographic origin, a significant differentiation was evident among all regions considering the whole sample (*F*_ST_ = 0.016, *p* < 0.001) and one copy per MLG (*F*_ST_ = 0.005, *p* < 0.001) (Table 5), although this was not observed when Cap Bon and Jendouba are compared (*F*_ST_ = 0.002, *p* = 0.098) (Table 5(B)).

The global analysis of differentiation by season also showed significant differences using the entire dataset (*F*_ST_ = 0.022, *p* < 0.001) and considering one copy per MLG (*F*_ST_ = 0.008, *p* < 0.001). Pairwise comparisons between autumn and spring/summer showed significant differences, but not between autumn and winter (Table 6).

The analysis for samples pooled by year of sampling showed a global significant differentiation when both the whole dataset (*F*_ST_ = 0.015, *p* < 0.001) and one copy per MLG (*F*_ST_ = 0.010, *p* < 0.001) were used (Table 7). Only the pairwise comparison of samples from 2011 and 2012 showed a non-significant difference (Table 7(B)).

Regarding genetic differences of samples according to the host plant, it was revealed that aphids collected on peach and potato were weakly but significantly differentiated, both when considering the whole sample (*F*_ST_ = 0.038, *p* < 0.001) or excluding copies (*F*_ST_ = 0.018, *p* < 0.001). This reveals that some MLGs were differentially represented on a specific host based on their frequencies.

### 3.5. Genetic Differentiation Based on the Molecular Differences Among MLGs

The genetic structuring based on hierarchical AMOVA revealed very low and significant genetic differentiation among and within *M. persicae* for both analyses performed by including clonal copies (*F*_ST_ = 0.07, *p* < 0.001) or with one representative copy per genotype (*F*_ST_ = 0.038, *p* < 0.001), respectively (Table 8(A,B)) most variation, 78.9%, being within individuals of the 19 populations followed by 17.3% of variation among individuals within populations and the remaining 3.8% among populations (Table 8(B)). By pooling *M. persicae* samples according to the sampling sites, season, year and host plant categories, unstructured populations were still observed thus 79.2%, 79.0%, 79.1% and 78.3% of variation was found between individuals respectively however, only about 0.4%, 0.9%, 1.0% and 1.8% of variation was recorded among populations of the main sources of variation (Table 8(C–F)).

Interestingly, the genetic clustering evidence that MLGs sampled from peach and potato formed different genetic groups denoted from A to I (ranging from 25 to 75 MLGs) with the frequency of unique MLGs dominant over groups. In light of this fact, it is worth noting that all groups comprising unique genotypes from potato, peach and common genotypes except group “I” showed 25 unique genotypes, which implies that the tree is constituted by mixed genotypes from both hosts (Figure 3).

### 3.6. Genetic Differentiation Based on a Bayesian Approach

The Bayesian clustering analysis using an admixture model revealed that the most probable number of genetic clusters was K = 3 (Ln P(K) = −8181.90, ΔK = 60.87) when considering only one copy per MLG (N = 397) (Figure 4A). The MLGs were assigned to a specific cluster when their coefficient of ancestry was higher to 0.7. Thus, clusters 1, 2 and 3 contained 34.5%, 34.8% and 25.2% of MLGs respectively, while 5.5% of MLGs were not assignable to any cluster. Clusters 1 and 2 had a predominant composition of MLGs sampled on peach (65.7% and 76.1%, respectively) than on potato (34.3% and 23.9%, respectively), while cluster 3 had almost an equal composition of MLGs from peach and potato (51% and 49%, respectively). When the whole dataset was used (all MLG copies), the Evanno’s approach failed to determine a single K with a high ΔK value (Appendix A).

When the analysis was performed on populations rather than on MLGs, the number of hypothetical populations was K = 2 (Ln P(K) = −8946.87, ΔK = 381.11) (Appendix A). Seven out of 10 populations sampled on peach grouped together in cluster 1 (pop 1, 3, 5, 6, 9, 12, 14, and 16), while cluster 2 mainly comprised samples from potato (pop 4, 7, 8, 10, 11, 13, 15 and 17–19). (Figure 4B)

The migration test between peach orchards and potato crops revealed lower frequencies of private alleles [p(1) = 0.012] and 2.8 migrants per generation after correcting for sampling size. Migration between geographic zones (Cap Bon, Jendouba and Kairouan) was even higher (p(1) = 0.005; 17.6 migrants).

## 4. Discussion

### 4.1. Clonal Diversity and Genetic Composition of M. persicae Populations in Tunisia

All of the six microsatellite loci studied were polymorphic and produced a total of 92 alleles (~15 alleles per locus in average), results which in terms of the level of polymorphism are comparable to those found for other *M. persicae* populations worldwide. For instance, a recent study on Italian populations of the peach potato aphid showed a total of 90 alleles at 8 microsatellite loci (~11 alleles per locus) [49]. Similarly, Chinese populations of the peach potato aphid studied at seven microsatellites revealed 174 alleles (~29 alleles per locus) [48], while a restricted sample of Tunisian *M. persicae* aphids revealed 49 alleles (~10 alleles per locus) at five microsatellite loci [22].

In terms of the genetic diversity, our study (2011–2013 and 2016) shows an overall clonal diversity (G/N = 0.72) similarly high to that previously reported (2005–2009; G/N = 0.78) for *M. persicae* from Tunisia [22]. Furthermore, Tunisian populations of the green peach aphid are featured by a high proportion of unique genotypes (90.7%), not far from the 86.7% previously reported by Charaabi et al. [22] When samples are grouped according to the host plant on which they were collected, the clonal diversity appears slightly higher on aphids sampled from peach orchards (G/N = 0.76) than from potato crops (G/N = 0.70). Clonal diversity for the peach potato aphid is known to depend on biotic and abiotic factors, among which the most determining are (i) the availability of peach trees (*P. persica*), the primary host where *M. persicae* reproduce sexually, and (ii) the harshness of the winter climate, and the management practices involved (e.g., insecticide applications). Parthenogenetic reproduction can quickly produce so-called ‘genetic inflation’, a population genetic phenomenon where a few asexual genotypes are overrepresented, while unique genotypes are rarely found [11]. For instance, lower clonal diversities are detected for *M. persicae* aphids collected on secondary hosts and weeds, on which *M. persicae* females multiply parthenogenetically during spring and summer. This has been reported for Scottish populations (G/N = 0.042), where most overrepresented genotypes were also resistant to insecticides [50] and confirmed in larger samples caught in suction traps (G/N = 0.014) [51]. Similarly, French peach potato aphids sampled from oilseed rape also show a low clonal diversity (G/N = 0.06) [52]. In contrast, peach potato aphids sampled from peach orchards and suction traps show larger clonal diversities, as reported in northern France (G/N = 0.69) [10]. Also, higher proportions of unique genotypes have been found in areas where peach is cultivated in Greece and southern Italy [53].

Regarding the impact of climate conditions on clonal diversity, *M. persicae* populations from areas with warmer winters tend to show lower clonal diversities than in areas with colder winters, as reported for countries from the northern and southern hemispheres. For example, in the southeast of Spain, where agroecosystems are quite heterogeneous comprising peach orchards surrounded by cultivated and wild herbaceous hosts, the clonal diversity is substantially lower (G/N = 0.46) than in Tunisia or France [54]. Similarly, in Australia (G/N = 0.44) [12], New Zealand (G/N = 0.32) [55] and Chile (G/N = 0.31) [56], clonal diversity is also lower, which is a common feature of recently introduced aphid populations in countries with warmer climates [13].

In our study, most aphids were sampled during their asexual phase (spring/summer), so clonal diversity is affected by genetic inflation [11]. However, the high clonal diversity together with a significant heterozygote deficiency and low LD strongly suggest that *M. persicae* in Tunisia mainly reproduce through CP. Nevertheless, two MLGs (G6 and G331) were overrepresented and time-persistent. Genotype G6 was the most abundant genotype over the whole sample (29 clonal copies on peach and 1 copy on potato), while genotype G331, the only one found through the all sampled years (from 2011–2013 and 2016), was restricted to potato crops regardless of the season and the sampling sites (7 copies in winter 2011 at Jendouba; 1 copy in autumn 2012 at Cap Bon; 4 copies in winter 2013 at Jendouba; 5 copies in spring 2016 at Cap Bon). Whether these genotypes constitute ‘superclones’ [12,13] is something that needs to be further studied, including the study of insecticide resistance mechanisms, which is currently an on-going project.

### 4.2. Genetic Differentiation Among Populations

The global AMOVA analysis revealed a weak but significant genetic differentiation (*F*_ST_ = 0.038; one copy per genotype), as well as significant pairwise *F*_ST_ comparisons among the 19 populations of *M. persicae*, ranging from 0.013–0.083, with most (99%) variation found within each sampling localities. Our results are similar to those previously reported by Charaabi et al. [22] for peach potato aphid populations in Tunisia, who also found a weak population differentiation (*F*_ST_ = 0.024) with most variation (94%) present within localities. These findings suggest that the predominance of sexual reproduction explains the genetic homogeneity among populations, as aphids migrate from different hosts to peach trees where they mate, involving recombination of the available population [10]. The presence of peach orchards in all of the sampled localities (Cap Bon, Jendouba and Kairouan) agrees with this view. In addition, the weak genetic differentiation among peach potato aphid populations can be explained due to sampled populations being subjected to similar selection forces, including climate, host plant, and management practices, and/or due to the lack of barriers to the gene flow [6].

The life cycle of *M. persicae* is closely dependent on climate, as sexual reproduction produces cold-resistant eggs, while obligate parthenogenetic individuals reproduce asexually all-year-round when winters are mild, a biological feature that has previously been posited to explain aphid population structures, e.g., [57,58]. In our study, the sampled localities share some climate features, such as the mean minimum of temperature (~13 °C) and the mean maximum of temperature (~25 °C) [16]. Besides the climatic conditions, the geographic distance between peach and potato plots within a sampling location ranged between some 1 to 20 km, a distance easily allowing winged aphids to move and circulate freely between peach orchards and potato crops. This view is supported by the low frequency of private alleles (0.012), the low proportion of variation among the primary and secondary hosts (1.8%) and the high variation within individuals among populations (78.9%), results similar to those previously found for peach potato populations [48,54]. Our results support the contention that neither climate nor host is acting as a significant selective agent on genetic diversity in peach potato aphid populations in Tunisia, hence that sexuality combines alleles from aphids coming from different hosts and thus homogenizes the genetic diversity, as earlier reported [59].

That the peach potato aphid has become a cosmopolitan pest in part due to its ability to move over large geographic distances. Thus, it has colonized all continents through clonal propagation, for which human transportation and global commerce have aided in expanding its range both at global and regional scales [9,60]. Although physical barriers, like rivers and mountains, may reduce inter-population gene flow as observed in populations of *M. persicae* from China [48], individuals can be airborne for 50–100 km [13,61], as they produce winged morphs during the sexual and asexual phases. Hence, the lack of spatial genetic differentiation observed in Tunisia can also be explained by the absence of significant geographical barriers between the sampling locations (Cap Bon, Jendouba and Kairouan), which are separated by 100–150 km. Similar unstructured and weakly differentiated spatial patterns of the genetic diversity have been observed in several *M. persicae* populations worldwide, including Australia (*F*st = 0.021) [12], Chile (*F*_ST_ = 0.012) [56] and Italy (*F*_ST_ = 0.137) [49]. Our study revealed very low variation among localities, both at analyzing data using all clonal copies (1.61%) or one copy per genotype (0.50%) this is due to a high gene flow between populations of the peach potato aphid [62], as suggested by the migration test that computed a high proportion of migrant individuals (17.59%) between the sampled localities in Tunisia. Interestingly, when the distribution of the commonest genotypes was studied at a geographic scale, genotype G6 (30 copies) and G247 (22 copies) were restricted to the Cap Bon region, which is reminiscent of the Scottish clone “L” that showed a strong invasion to northeast Scotland, for which the authors argued was due to geographic restrictions [51], although this need to be confirmed for the Tunisian situation.

Temporal dynamics of the genetic diversity in populations of *M. persicae* can be governed by positive selection exerted by seasonal changes related to variable biotic and abiotic selective regimes, such as host plant availability, abundance of natural enemies, weather conditions, and management practices [63]. Our study revealed a temporal shift in the frequency of the commonest genotypes G370, G331 and G159, which changed from 17, 7 and 2 copies to 1, 5 and 3 copies respectively, between 2011 and 2016. For instance, a recent study has proposed that fluctuations in population densities of *M. persicae* are positively correlated with the maximum temperatures and with the presence of natural enemies (e.g., parasitoid wasps, ladybirds, staphylinids and spiders) and negatively correlated with the relative humidity during the evening [64]. On the other hand, the globalization of agriculture particularly the application of insecticides, is playing a substantial role in shaping aphid populations, as insecticides constitute a strong selection agent and confer aphid individuals carrying resistance mechanisms, and/or metabolic resistance related to insensitivity to the compounds used [65]. In the case of *M. persicae*, insecticide resistance has accelerated the widespread distribution of this aphid pest, even favoring some genotypes being shared in different countries [6,52]. The dynamic of insecticide resistance of Tunisian populations of *M. persicae* is variable according to regions, hosts and season, as previously reported [22]. Thus, those overrepresented genotypes of *M. persicae* in Tunisian populations and their time-persistence among seasons and years could be attributed to selection by insecticides. Further research is needed to confirm this and understand how insecticide resistance is evolving in Tunisian populations.

The population dynamics of *M. persicae* is important for transmission efficiency of phytopathogenic viruses. This is because the affinity for some viruses is variable among different *M. persicae* genotypes; some *M. persicae* genotypes transmit the virus in a non-persistent way while others are persistent, as described for PVY and PLRV, respectively [66]. However, depending on its efficiency as vector (poor, intermediate or efficient), some *M. persicae* genotypes may increase virus transmission [67]. Several aphids are implicated in the transmission of plum pox virus PPV [68]. Although *M. persicae* is comparatively less important in the spread of PPV, due to the fact that during the period when peach orchards are more susceptible the abundance of *M. persicae* is lower, it has been shown that *M. persicae* is highly efficient in PPV transmission [69]. This emphasizes the relevance of knowing the genetic characteristics of *M. persicae* for adopting pre-emptive rational control measures.

## 5. Conclusions: Contribution to the Control Management of the Peach Potato Aphid

In Tunisia, about 20,000 tons of seed potatoes are imported annually from France and The Netherlands [70], from which 10% is assigned to produce local certified seeds. One of the main challenges for the Tunisian agriculture is ensuring virus-free seeds, as the estimations of yield losses are around 35% caused by PVY and 30% by PLRV, viruses transmitted by *M. persicae* [20,71]. By using high-resolution genetic markers, as here, we studied the genetic structure of *M. persicae* populations in Tunisia, in order to increase our understanding of the factors shaping populations in order to predict their evolutionary responses to the present management practices. As the peach potato aphid is under strong selection pressures (e.g., insecticides), this will certainly result in the selection of resistant genotypes. Thus, the evolution of genetic diversity and population structure in time and space could be faster, making pest control harder and more expensive. Our results should certainly be considered, we believe, when considering novel management strategies that better fit with the biology of the pest, given that aphids take advantage of the oversimplified design of current agroecosystems and succeed in a range of environments. Following this survey, we are studying the temporal and spatial dynamics of insecticide resistance in the peach potato aphid, as key knowledge for improving its control in Tunisia.

## Figures and Tables

**Figure 1 insects-10-00330-f001:**
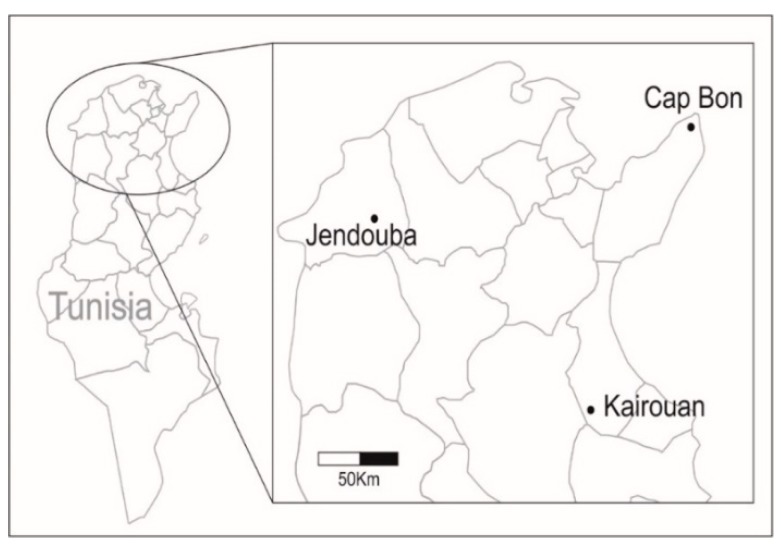
Localities in Tunisia sampled for *M. persicae*. Peach orchards and potato crops located in Jendouba, Cap Bon and Kairouan were monitored from 2011 to 2016.

**Figure 2 insects-10-00330-f002:**
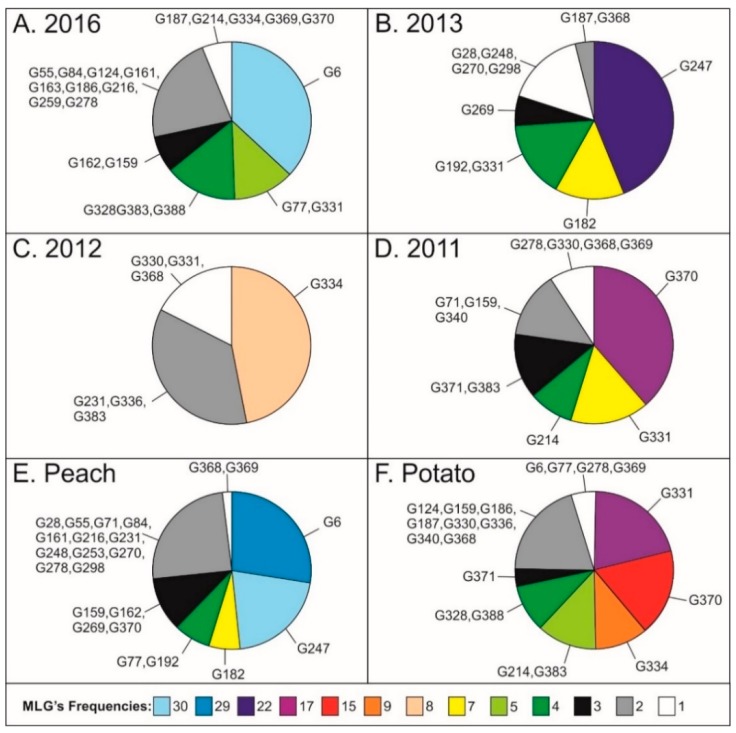
Distribution of the commonest genotypes of *M. persicae* according to (**A**–**D**) the sampling year; (**E**,**F**) host plant.

**Figure 3 insects-10-00330-f003:**
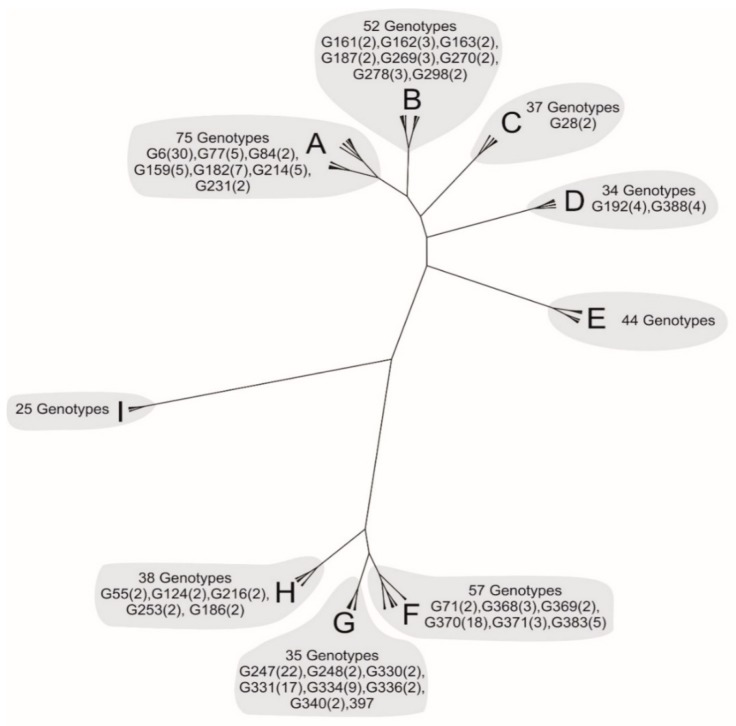
Neighbor-joining tree showing the genetic relatedness among the 397 MLGs found on *M. persicae* populations from Tunisia. Sampled individuals were genotyped at six microsatellite loci and the tree was built based on allele shared distance (DAS method). The total number of genotypes within each group is shown, but only multicopy MLGs are cited with their frequencies in parentheses. The genetic distance among MLGs was computed considering a bootstrap of 1000 replicates.

**Figure 4 insects-10-00330-f004:**
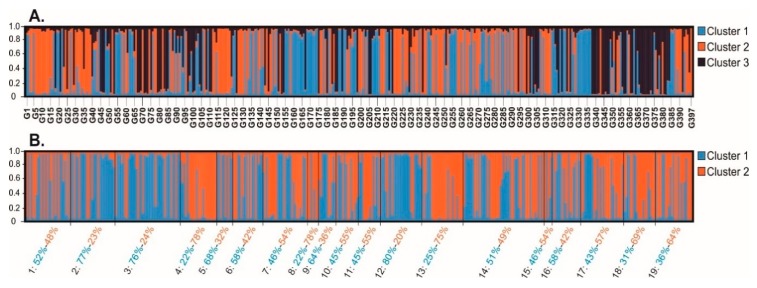
Assessment of population genetic structure by Bayesian cluster analysis of *M. persicae* in Tunisia. The clustering analysis was performed using an admixture model on (been referred) (**A**) 397 MLGs (assuming k = 3; see Materials and Methods and Appendix A); (**B**) 19 populations using one copy per MLG (assuming k = 2). The MLGs or populations are represented by vertical bars divided by colors according to their coefficients of ancestry to each cluster.

**Table 1 insects-10-00330-t001:** Number of *M. persicae* collected on peach orchards and potato seed crops in three localities in Tunisia during 4 years of monitoring. Samples were grouped on 19 populations according to the host, season, locality and year of collection. The number of individuals examined is shown in parentheses.

	Potato			Peach		
Locality	Spring	Autumn	Winter	Spring	Autumn	Winter
Cap Bon						
2016	pop1 (37)			pop2 (64)		
2013				pop3 (77)		
2012		pop4 (24)			pop5 (10)	
2011			pop6 (23)			pop7 (28)
Jendouba						
2016	pop8 (8)			pop9 (15)		
2013			pop10 (17)			
2012			pop11 (22)			pop12 (26)
2011			pop13 (47)			pop14 (51)
Kairouan						
2016	pop15 (5)			pop16 (20)		
2012						pop17 (28)
2011			pop18 (21)			pop19 (25)
Subtotal	204			344		
Total aphids	548					

**Table 2 insects-10-00330-t002:** Genetic diversity indexes of *M. persicae* populations collected from peach and potato in Tunisia. Population (Pop), number of individuals analyzed (N), total number of multilocus genotypes (G), clonal diversity (G/N), ratio of unique/multicopy genotypes (U/M), Shannon diversity index (H) and its evenness (VH), Simpson diversity index (D) and its evenness (ED), mean number of alleles (Na), allelic richness over loci (A), mean expected heterozygosity (He), observed heterozygosity (Ho), loci under disequilibrium out of possible tests (LD), inbreeding coefficient (*F*_IS_) considering all clonal copies or one single copy per genotype.

																	*F* _IS_		
Pop	Host	Locality	Season	N	G	G/N	U/M	H	VH	D	ED	Na	A	He	Ho	LD	All Copies	One Copy	*p*-Value
1	potato	Cap Bon	spring	37	28	0.757	25/3	3.117	0.946	0.970	0.727	7.500	4.74	0.781	0.548	8/15	0.269	0.303	<0.01
4	potato	Cap Bon	autumn	24	23	0.958	22/1	3.120	0.995	0.996	0.000	7.000	4.62	0.779	0.739	6/15	0.052	0.053	<0.01
6	potato	Cap Bon	winter	23	19	0.826	17/2	2.834	0.963	0.972	0.396	8.833	4.92	0.770	0.631	10/15	0.145	0.184	<0.01
8	potato	Jendouba	spring	8	7	0.875	6/1	1.906	0.979	0.964	0.000	4.500	4.11	0.723	0.667	2/15	0.082	0.084	NS
10	potato	Jendouba	winter	17	14	0.823	13/1	2.507	0.949	0.956	0.000	5.833	4.50	0.781	0.643	5/15	0.150	0.183	<0.01
11	potato	Jendouba	winter	22	14	0.636	12/2	2.272	0.861	0.874	0.235	8.167	5.52	0.828	0.714	3/15	0.087	0.141	<0.01
13	potato	Jendouba	winter	47	25	0.532	21/4	2.675	0.831	0.893	0.590	6.833	4.48	0.758	0.647	9/15	0.009	0.148	<0.01
15	potato	Kairouan	spring	5	5	1.000	5/0	1.609	1.000	1.000	−1.000	4.667	4.67	0.767	0.567	2/15	0.284	0.284	<0.01
18	potato	Kairouan	winter	21	20	0.952	19/1	2.978	0.994	0.995	0.000	8.667	4.98	0.785	0.617	5/15	0.208	0.219	<0.01
**Subtotal**				**204**	**142**	**0.696**	**126/16**	**4.533**	**0.915**	**0.983**	**0.914**			**0.798**	**0.641**	**12/15**			
2	peach	Cap Bon	spring	64	28	0.437	22/6	2.430	0.729	0.793	0.397	7.000	4.33	0.736	0.744	13/15	−0.253	−1.011	<0.01
3	peach	Cap Bon	spring	77	41	0.532	33/8	3.097	0.834	0.909	0.634	8.833	4.83	0.782	0.581	12/15	0.170	0.259	<0.01
5	peach	Cap Bon	autumn	10	10	1.000	10/0	2.302	1.000	1.000	−1.000	5.500	4.42	0.749	0.533	1/15	0.299	0.299	<0.01
7	peach	Cap Bon	winter	28	28	1.000	28/0	3.332	1.000	1.000	−1.000	7.500	4.57	0.745	0.571	1/15	0.237	0.237	<0.01
9	peach	Jendouba	spring	15	11	0.733	8/3	2.303	0.961	0.952	0.687	5.667	4.42	0.758	0.773	12/15	−0.057	−0.020	NS
12	peach	Jendouba	winter	26	26	1.000	26/0	3.258	1.000	1.000	−1.000	8.333	4.51	0.731	0.487	3/15	0.338	0.338	<0.01
14	peach	Jendouba	winter	51	51	1.000	51/0	3.932	0.999	1.000	−1.000	8.333	4.51	0.746	0.614	4/15	0.178	0.178	<0.01
16	peach	Kairouan	spring	20	18	0.900	16/2	2.857	0.988	0.989	0.529	6.667	4.43	0.745	0.555	6/15	0.241	0.260	<0.01
17	peach	Kairouan	winter	28	27	0.964	26/1	3.283	0.996	0.997	0.000	7.000	4.41	0.745	0.660	2/15	0.115	0.115	<0.01
19	peach	Kairouan	winter	25	23	0.920	22/1	3.087	0.984	0.990	0.000	7.000	4.38	0.752	0.688	0/15	0.065	0.086	NS
**Subtotal**				**344**	**262**	**0.762**	**240/22**	**5.196**	**0.933**	**0.988**	**0.808**			**0.775**	**0.617**	**10/15**			
**Whole Sample**				**548**	**397**	**0.724**	**360/37**	**5.585**	**0.933**	**0.992**	**0.912**								**<0.001**

**Table 3 insects-10-00330-t003:** Frequency of each multicopy genotype in the whole sample and according to the host plant where *M. persicae* aphids were sampled.

Genotype	Number of Sampled Aphids	Frequency		
		Whole Sample	Peach	Potato
G6	30	0.054	0.084	0.005
G247	22	0.040	0.064	-
G370	18	0.033	0.009	0.073
G331	17	0.031	-	0.083
G334	9	0.016	-	0.044
G182	7	0.013	0.020	-
G77	5	0.091	0.012	0.005
G159	5	0.091	0.009	0.010
G214/G383	5	0.091	-	0.024
G192	4	0.073	0.012	-
G328/G388	4	0.073	-	0.020
G162/G269	3	0.005	0.009	-
G278	3	0.005	0.006	0.005
G368	3	0.005	0.003	0.010
G371	3	0.005	-	0.015
G28/G55/G71/G84/G161/G163/G216/G231/G248/G253/G270/G298	2	0.004	0.006	-
G124/G186/G187/G330/G336/G340	2	0.004	-	0.010
G369	2	0.004	0.003	0.005
16 multicopy genotypes from potato	78	0.142	-	0.382
22 multicopy genotypes from peach	104	0.190	0.302	-
37 multicopy genotypes over the whole sample	188	0.343		

**Table 4 insects-10-00330-t004:** Pairwise comparisons of fixation index (*F*_ST_) between each pair of populations. Only one copy per multilocus genotype (MLG) was considered for computing the *F*_ST_ values.

	pop1	pop2	pop3	pop4	pop5	pop6	pop7	pop8	pop9	pop10	pop11	pop12	pop13	pop14	pop15	pop16	pop17	pop18	pop19
**pop1**																			
**pop2**	0.050																		
**pop3**	0.049	0.044																	
**pop4**	0.040	0.072	0.060																
**pop5**	0.056	0.032	0.036	0.054															
**pop6**	0.034	0.070	0.036	0.018	0.039														
**pop7**	0.038	0.041	0.035	0.028	0.024 ^NS^	0.001 ^NS^													
**pop8**	0.020 ^NS^	0.075	0.053	0.001 ^NS^	0.063	0.015 ^NS^	0.018 ^NS^												
**pop9**	0.076	0.071	0.046	0.060	0.077	0.061	0.060	0.080											
**pop10**	0.027 ^NS^	0.067	0.035	0.009 ^NS^	0.057	0.024 ^NS^	0.043	0.000 ^NS^	0.053										
**pop11**	0.037	0.075	0.041	0.020 ^NS^	0.071	0.029	0.052	0.008 ^NS^	0.060	0.004 ^NS^									
**pop12**	0.055	0.041	0.045	0.074	0.043	0.035	0.031	0.069	0.080	0.054	0.066								
**pop13**	0.064	0.082	0.081	0.002 ^NS^	0.054	0.039	0.044	0.021 ^NS^	0.097	0.033	0.037	0.092							
**pop14**	0.039	0.034	0.030	0.037	0.026	0.040	0.013	0.035	0.068	0.038	0.060	0.052	0.054						
**pop15**	0.044 ^NS^	0.048	0.054 ^NS^	0.014 ^NS^	0.003 ^NS^	0.014 ^NS^	0.013 ^NS^	0.034 ^NS^	0.079	0.027	0.040 ^NS^	0.046 ^NS^	0.007 ^NS^	0.035 ^NS^					
**pop16**	0.050	0.049	0.022	0.050	0.066	0.036	0.035	0.054	0.018 ^NS^	0.044 ^NS^	0.066	0.070	0.083	0.035	0.063 ^NS^				
**pop17**	0.055	0.043	0.034	0.030	0.054	0.032	0.021	0.057	0.026	0.042	0.059	0.054	0.064	0.027	0.050	0.010 ^NS^			
**pop18**	0.046	0.077	0.036	0.020	0.056	0.018 ^NS^	0.025	−0.000 ^NS^	0.047	0.014 ^NS^	0.006 ^NS^	0.052	0.045	0.049	0.044 ^NS^	0.047	0.039		
**pop19**	0.027	0.050	0.034	0.007 ^NS^	0.046	0.016 ^NS^	0.016 ^NS^	−0.004 ^NS^	0.059	0.010 ^NS^	0.023	0.055	0.036	0.019	0.032 ^NS^	0.033	0.016	0.010 ^NS^	

Significant *F*_ST_ values are shown below diagonal (*p* < 0.05); ^NS^: non-significant.

**Table 5 insects-10-00330-t005:** Assessment of the genetic differentiation by the geographic origin in *M. persicae* samples. The fixation index *Fst* was computed using (**A**) all clonal copies per multilocus genotype or (**B**) one single copy.

(A)	Cap Bon	Jendouba	Kairouan	(B)	Cap Bon	Jendouba	Kairouan
Cap Bon	-			Cap Bon	-		
Jendouba	0.017 *	-		Jendouba	0.002	-	
Kairouan	0.019 *	0.010 *	-	Kairouan	0.007 *	0.008 *	-

* (*p* < 0.001).

**Table 6 insects-10-00330-t006:** Assessment of the genetic differentiation by the season sampled in *M. persicae* aphids. The fixation index *Fst* was computed using (**A**) all clonal copies per multilocus genotype or (**B**) one single copy.

(A)	Autumn	Spring	Winter	(B)	Autumn	Spring	Winter
Autumn	-			Autumn	-		
Spring	0.033 *	-		Spring	0.020 *	-	
Winter	0.004	0.024 *	-	Winter	0.005	0.009 *	-

* (*p* < 0.001).

**Table 7 insects-10-00330-t007:** Assessment of the genetic differentiation by the year of sampling in *M. persicae* aphids. The fixation index *Fst* was computed using (**A**) all clonal copies per multilocus genotype or (**B**) one single copy.

(A)	2016	2013	2012	2011	(B)	2016	2013	2012	2011
2016	-				2016	-			
2013	0.024 *	-			2013	0.015 *	-		
2012	0.005 *	0.040 *	-		2012	0.008 *	0.015 *	-	
2011	0.008 *	0.051 *	0.009 *	-	2011	0.012 *	0.018 *	0.004	-

* (*p* < 0.001).

**Table 8 insects-10-00330-t008:** Hierarchical analyses of the molecular variance AMOVA for *M. persicae* aphids from Tunisia. df: Degrees of freedom.

Source of Variations	df	Sum of	Variance	Percentage	Fixation	*p-*Value
		Squares	Components	Variation	Indices	
**(A) 19 Populations (all data)**						
Among populations	18	219.954	0.17294	7.22	*F*_ST_: 0.07225	0.00000
Among individuals within populations	529	1305.616	0.24727	10.33	*F*_IS_: 0.11134	0.00000
Within individuals	548	1081.500	1.97354	82.44	*F*_IT_: 0.17555	0.00000
**(B) 19 Populations (single copy per genotype)**						
Among populations	18	119.093	0.08998	3.78	*F*_ST_: 0.03781	0.00000
Among individuals within populations	400	1081.083	0.41281	17.34	*F*_IS_: 0.18027	0.00000
Within individuals	419	786.500	1.87709	78.87	*F*_IT_: 0.21127	0.00000
**(C) *M. persicae* Pooled According to Geographical Origin (single copy per genotype)**						
Among populations	2	11.359	0.01049	0.44	*F*_ST_: 0.00441	0.02151
Among individuals within populations	414	1180.714	0.48474	20.39	*F*_IS_: 0.20477	0.00000
Within individuals	417	785.000	1.88249	79.17	*F*_IT:_ 0.20828	0.00000
**(D) *M. persicae* Pooled According to Season of Sampling (single copy per genotype)**						
Among populations	2	14.846	0.02053	0.86	*F*_ST_: 0.00861	0.00000
Among individuals within populations	414	1177.227	0.48052	20.16	*F*_IS_: 0.20335	0.00000
Within individuals	417	785.000	1.88249	78.98	*F*_IT_: 0.21021	0.00000
**(E) *M. persicae* Pooled According Year of Sampling (single copy per genotype)**						
Among populations	3	22.920	0.02424	1.02	*F*_ST_: 0.01018	0.00000
Among individuals within populations	413	1169.153	0.47419	19.92	*F*_IS_: 0.20121	0.00000
Within individuals	417	785.000	1.88249	79.07	*F*_IT_: 0.20934	0.00000
**(F) *M. persicae* Pooled According to Host Plant (single copy per genotype)**						
Among populations	1	18.553	0.04269	1.78	*F*_ST_: 0.01784	0.00000
Among individuals within populations	402	1136.309	0.47582	19.88	*F*_IS_: 0.20241	0.00000
Within individuals	404	757.500	1.87500	78.34	*F*_IT_: 0.21663	0.00000

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
