# Peer review of "Spatial and Temporal Genetic Diversity of the Peach Potato Aphid Myzus persicae (Sulzer) in Tunisia"

_insects, 2019, doi:10.3390/insects10100330_

Round 1

Reviewer 1 Report

The research is well designed and carried out. The techniques used and the computer programs used are interesting. There are some small revisions to be made to the text (see pdf).

Basically, I cannot understand what the practical effect of this (well executed, conducted and explained) research on the control of Myzus persicae as a virus vector could be. But the fault lies in my poor visionary ability.

The paper can be published after a few minor revision.

Reviewer 2 Report

The manuscript submitted by Hlaoui and colleagues investigates (using a microsatellite-based approach) the genetic characteristics of Myzus persicae populations in Tunisia to understand the distribution of aphid genotypes and the spread/relevance of clonality in M. persicae populations. As a whole, Authors assessed that sampled populations are grouped on the basis of their host plants and that M. persicae reproduces mainly through cyclical parthenogenesis in Tunisia.

The experimental plan is well organized, data amount is sufficient and the description of the methods is appropriate. 

I have no major concerns about the structure of the paper neither about data that are clearly reported, whereas I think that the discussion could be improved. In particular I think that Author should focus only on the genetic differentiation among populations, whereas several sections about insecticide resistance and virus transmission should be deleted not only because M. persicae is scarcely important in the spread of PPV, but also since Authors’ statements are true if the observed clones in Tunisia have different vectoriality/resistance (and these data are not available). Similarly, conclusion section is really vague. Authors stated several times in their text that their data will be useful of the aphid management but actually they fail in suggesting a true application.

As a whole, I think that the present version of the manuscript should be accepted after moderate revision related to discussion and conclusion. Lastly, even if the manuscript is generally well written, I think that the paper would benefit from proof-reading by a native English speaker or professional proof-reading service.

Reviewer 3 Report

The authors present a study on spatial and temporal genetic diversity of 19 M. persicae populations on two hosts basing on 6 SSRs. 

Basing on the dataset extensive analysis of population genetic measures were performed (all but Psex values) and also all presented. The main results: cyclically parthenogensis of M. persicae populations in Tunisia and low genetic differentiation among populations is thus documented in many (redudandant) ways. 

The authors try to input novelty by discussing the effects of the (non-existing genetic structure) on abiotic and biotic stresses & virus transmission, but do not provide deep insights or ideas. 

My conclusion, a solid piece of scientific work to tediously document what may be computed with molecular marker data - but yet not very innovative to the reader.

The abstract must be reviewed. 

Some particular points: 

l19 not genetic features, you studied the genetic structure l32ff where is that farm-scale study elaborated and discussed pertaining virus vectors?  table 37 MLGs?

Reviewer 4 Report

See attached comments to authors.
